# Obtaining Information about Operation of Centrifugal Compressor from Pressure by Combining EEMD and IMFE

**DOI:** 10.3390/e22040424

**Published:** 2020-04-09

**Authors:** Yan Liu, Kai Ma, Hao He, Kuan Gao

**Affiliations:** School of Mechanical Engineering, Northwestern Polytechnical University, Xi’an 710072, China; makai@mail.nwpu.edu.cn (K.M.); haohe@mail.nwpu.edu.cn (H.H.); 15991737832@mail.nwpu.edu.cn (K.G.)

**Keywords:** centrifugal compressor, surge, nonlinear dynamics, multi-scale fuzzy entropy, ensemble empirical mode decomposition

## Abstract

Based on entropy characteristics, some complex nonlinear dynamics of the dynamic pressure at the outlet of a centrifugal compressor are analyzed, as the centrifugal compressor operates in a stable and unstable state. First, the 800-kW centrifugal compressor is tested to gather the time sequence of dynamic pressure at the outlet by controlling the opening of the anti-surge valve at the outlet, and both the stable and unstable states are tested. Then, multi-scale fuzzy entropy and an improved method are introduced to analyze the gathered time sequence of dynamic pressure. Furthermore, the decomposed signals of dynamic pressure are obtained using ensemble empirical mode decomposition (EEMD), and are decomposed into six intrinsic mode functions and one residual signal, and the intrinsic mode functions with large correlation coefficients in the frequency domain are used to calculate the improved multi-scale fuzzy entropy (IMFE). Finally, the statistical reliability of the method is studied by modifying the original data. After analysis of the relationships between the dynamic pressure and entropy characteristics, some important intrinsic dynamics are captured. The entropy becomes the largest in the stable state, but decreases rapidly with the deepening of the unstable state, and it becomes the smallest in the surge. Compared with multi-scale fuzzy entropy, the curve of the improved method is smoother and could show the change of entropy exactly under different scale factors. For the decomposed signals, the unstable state is captured clearly for higher order intrinsic mode functions and residual signals, while the unstable state is not apparent for lower order intrinsic mode functions. In conclusion, it can be observed that the proposed method can be used to accurately identify the unstable states of a centrifugal compressor in real-time fault diagnosis.

## 1. Introduction

In recent years, centrifugal compressors have been used widely in industry. As an intrinsic characteristic of the centrifugal compressor, surge can cause flow-induced vibration and lower aerodynamic performance. Therefore, surge may increase the energy consumption of a centrifugal compressor and lead to fatigue of the structure or instability of aeroelasticity as the surge becomes violent [1,2].

Researchers have completed a lot of studies on surge, as energy-saving is required and centrifugal compressors are used in variable working conditions. Roughly speaking, there are two ways to study surge, namely, mechanism studies and identification from the data. In mechanism studies, it is believed that an external factor and two internal factors may induce surge. The external factor is high-pressure gas stored in the pipes, and the internal factors are rotation stall in the passage and separation of flow around the blades. Because the dynamic pressure at the outlet of a centrifugal compressor can indicate the state of the compressor accurately and immediately, it provides a way to study surge. In particular, the dynamic pressure becomes complex and seriously disordered as the centrifugal compressor undergoes transition from the stable state to unstable states, and routine methods, such as spectral analysis and time-domain analysis, fail to capture and describe this complex behavior exactly.

With the development of nonlinear dynamics, various nonlinear analysis methods have been widely applied in the surge prediction of centrifugal compressors, such as wavelet analysis, Lyapunov exponent, and fractal dimension [3,4,5,6]. The studies show that the flow from the inlet, impeller, and to the outlet behaves chaotically or disorderly as the working state of a centrifugal compressor is unstable. That is, the time sequence of the flow pressure becomes singular in the unstable state, and therefore some intrinsic properties should exist in the time sequence. Fortunately, entropy can be used to describe and measure such a singular phenomenon. Indeed, information entropy has been introduced to characterize dynamical systems between purely random, chaotic, and regular evolution [7,8,9]. In 2008, Chen and his group proposed the concept of fuzzy entropy (FE) and used the method to extract the characteristic information of a surface electromyogram signal [10,11]. Because of the many advantages of FE, it is widely used in economic management, medical diagnosis, weather forecasting, biology, ecology, and other fields [12,13]. Since then, some other methods are developed on the basis of entropy. Among them, multi-scale fuzzy entropy (MFE), as one method of the fuzzy entropy family, has advantages in analyzing a dynamical system by placing the sample data on different scales [14,15,16,17]. Similarly, ensemble empirical mode decomposition (EEMD) is also powerful in dealing with non-stationary and nonlinear data. Based on the scale feature of the signal itself, EEMD could decompose a signal into several intrinsic mode functions, in order to achieve better analysis performance [18]. In particular, it is suitable for analyzing nonlinear and non-stationary signal sequences because of its high signal-to-noise ratio. Hence, a method combining EEMD and MFE can be considered as a strategy to capture the intrinsic properties of dynamic pressure with different orders, especially for the dynamic pressure as a centrifugal compressor operates in an unstable state [19].

In this study, some fundamental theories related to MFE, improved multi-scale fuzzy entropy (IMFE), and EEMD, are used and developed to describe the dynamic pressure of flow at the outlet of a centrifugal compressor. Then, a method combining IMFE with EEMD is proposed to measure the complexity of the intrinsic mode function of dynamic pressure under different scale factors and the probability of new information with the changes of dimension. Finally, some conclusions are obtained, and the feasibility of the method is verified to identify surge.

## 2. Fundamental Theories

### 2.1. Multi-Scale Fuzzy Entropy and Improved Multi-Scale Fuzzy Entropy

Entropy is a concept that originated from the field of thermodynamics, and Shannon first applied entropy to the field of information theory and proposed the concept of information entropy to measure the uncertainty of an event [20]. Since then, the concept of entropy has gradually been generalized. According to functions and applications, entropy can be divided into approximate entropy, sample entropy, fuzzy entropy, and so on. Among these methods, FE can describe clearly the edge of adjacent classes of sample data, which is a weakness of information entropy, approximate entropy, and sample entropy. In particular, FE could accurately describe the complexity of a system, because of its strong consistency and reduced dependence on the length of data. With the advantages of small deviation, good continuity, free parameter selection, and strong anti-interference ability, FE is suitable for studying dynamic pressure at the outlet of a centrifugal compressor [8]. Multi-scale fuzzy entropy, as a special kind of fuzzy entropy, gives a measurement of the complexity of a sequence under different scale factors [15]. Hence, both MFE and IMFE are used in the paper to investigate the complex nonlinear dynamic characteristics of dynamic pressure at the outlet of a centrifugal compressor, and identify the unstable state of the system.

For time sequence *x*, FE, MFE and IMFE can be calculated as follows [21,22,23],

1. For an *N*-length time sequence *x*(*i*), *i* = (1, 2,…, *N*), construct *m*-dimension vector for *x*(*i*), with *m* being the embedded dimension,
(1)$$X_{j}^{m}=\{x(j),x(j+1),\ldots ,x(j+m-1)\}-x_{0}(j)\;,\;1\leq j\leq N-m+1$$
where
(2)$$x_{0}(j)=\frac{1}{m}\sum_{k=0}^{m-1}x(j+k)$$

2. Define the distance $d_{jl}^{m}$ between $X_{j}^{m}$ and $X_{l}^{m}$ as the maximum value of the difference between the two elements,
(3)$$\begin{array}{ll} d_{jl}^{m}=d\left[X_{j}^{m},X_{l}^{m}\right]= & \underset{k\in (0,m-1)}{\max}\{|x(j+k)-x_{0}(j)|-\left|x(l+k)-x_{0}(l)\right|\}\\ & 1\leq j\leq N-m+1,\;1\leq l\leq N-m+1,j\neq l \end{array}$$

3. Calculate the similarity of $d_{jl}^{m}$ by selecting the exponential function $\exp(-\ln(2)\cdot {\left(d_{jl}^{m}/r\right)}^{n})$ as a fuzzy function, where *n* and *r* are the gradient and width of the boundary, respectively. The similarity of $\Phi^{m}$ can be defined as follows,
(4)$$\Phi^{m}(n,r)=\frac{1}{N-m+1}\sum_{j=1}^{N-m+1}\frac{1}{N-m}\sum_{l=1,l\neq j}^{N-m+1}\exp(-\ln(2)\cdot {\left(d_{jl}^{m}/r\right)}^{n})$$

4. Repeat Equations (1)–(4), for obtaining *m* + 1 dimensional similarity, and $\Phi^{m+1}$ can be described.

5. Fuzzy entropy is defined as the negative natural logarithm of the ratio of $\Phi^{m}$ and $\Phi^{m+1}$, FE for a time sequence *x*(*i*) is shown in Equation (5),
(5)$$\mathrm{FE}(x(i),m,n,r)=-\ln(\frac{\Phi^{m+1}(n,r)}{\Phi^{m}(n,r)})$$

6. In order to calculate MFE, the coarse-grained data are obtained following Equation (6),
(6)$$y_{o}^{\tau }=\frac{1}{\tau }\sum_{i=(o-1)\tau +1}^{o\tau }x(i)\;,\;1\leq o\leq \frac{N}{\tau }$$
where *τ* is the scale factor. The coarse-grained data of MFE with two and three scale factors are shown in Figure 1.

Further, MFE for a time sequence *x*(*i*) is expressed as Equation (7),
(7)$$\mathrm{MFE}(x(i),m,n,r,\tau )=\mathrm{FE}(y_{o}^{\tau },m,n,r)$$

It should be noted that there are some shortcomings in MFE. First, MFE does not exactly correspond to the original time sequence in its complexity, and some characteristics in the original time sequence may be lost. Another shortcoming is the variability of the entropy results for high scale factors, because the time sequence becomes shorter as the scale factor increases. These may lead to an unstable measure of entropy.

To reduce these defects, Hamed proposed a new method named IMFE to obtain coarse-grained data, instead of the mean value method in MFE [23,24]. Coarse graining of improved multi-scale fuzzy entropy is expressed as Equation (7). The coarse-grained data of IMFE with two- and three-scale factors are shown in Figure 2. The differences between the data points and their corresponding averages become even more striking for the coarse-grained data of IMFE than those of MFE.
(8)$$z_{p}^{\tau }=\frac{1}{\tau }\sum_{i=p}^{p+\tau -1}x(i),\;1\leq p\leq N-\tau +1$$

According to Equation (8), the original time sequence is coarse-grained for IMFE, and IMFE for a time sequence *x*(*i*) can be obtained following Equation (9),
(9)$$\mathrm{IMFE}(x(i),m,n,r,\tau )=\mathrm{FE}(z_{p}^{\tau },m,n,r)$$

Subsequently, IMFE could be calculated, which can measure or describe the self-similarity and complexity of a time sequence under different scale factors. If the entropy of a sequence is larger than that of another sequence on most scales, the former is considered to be more complex than the latter. Moreover, if the entropy of a time sequence decreases monotonically with the increase of scale factor, it indicates that the structure of the time sequence is simple, and the main information is included in the entropy of the minimum scale factor.

From the definition of fuzzy entropy, it is clear that the calculation of fuzzy entropy is related to embedding dimension *m*, gradient *n*, and width *r* of the similar boundary tolerance, and the length *N* of data. For large values of embedding dimension *m*, a large number of data points (*N* = 10*^m^*–30*^m^*) is needed to obtain an accurate result. Therefore, the embedding dimension *m* is generally set at two initially, and more detailed information can be described with an increase of *m*, as the sequence is reconstructed. For the width of the boundary, it is vital to choose a suitable *r*. For large *r*, a lot of statistical data will be lost. For small *r*, the estimated statistical characteristics are unsatisfactory and the anti-noise capability becomes weak. Therefore, the range of *r* is generally selected to be from 0.1SD to 0.25SD, where SD is the standard deviation of the original data. After considering the characteristics of the dynamic pressure at outlet and the sample frequency, in this paper, the main parameters of IMFE are set as *r* = 0.15SD, *n* = 2, and *N* = 2048, respectively. Furthermore, in order to ensure that the value of fuzzy entropy is not affected by the length of coarse-grained data, the maximum of the scale factor *τ* is generally ten, since the length of the coarse-grained data is *N*/*τ* for data of length *N*.

### 2.2. Ensemble Empirical Mode Decomposition

Huang proposed a time-frequency analysis method named the Hilbert–Huang Transform (HHT), which is a method combining empirical mode decomposition (EMD) and the Hilbert transform [18]. EMD is a powerful tool used to analyze a nonlinear and non-flat signal, which can decompose the signal into a series of single component signals named intrinsic mode functions (IMFs). Every IMF represents an approximate simple harmonic oscillation function with a different frequency, and these IMF components contain all the frequency components of the original signal. However, there is modal aliasing in EMD, which will lead to reduced accuracy of IMF and the loss of physical significance of IMF. Following this, Wu and his colleagues proposed ensemble empirical mode decomposition (EEMD), on the basis of EMD [14]. Hence, the combination of EEMD and IMFE will be used to analyze the dynamic pressure at outlet of a centrifugal compressor.

The main steps of EEMD can be summarized as follows [18],

1. Add white noise *w*(*i*) to the original signal *x*(*i*),
(10)$$x_{q}(i)=x(i)+w_{q}(i),\;q=1,\ldots ,M$$
where $x_{q}(i)$ is the randomly shuffled signal at the *q*th time, and *M* is the maximum time.

2. Decompose $x_{q}(i)$ into *k* IMF components and a residual *R_q_*(*i*),
(11)$$x_{q}(i)=\sum_{j=1}^{k}c_{qj}(i)+R_{q}(i)$$
where *c_qj_*(*i*) is the *j*th IMF component of $x_{q}(i)$, it is calculated by using EMD method

3. In order to eliminate effects of multiple adding white noise on IMF components, the average values of the IMF components are calculated as follows. Then, the new IMF components *C_j_*(*i*) and the remainder *R*(*i*) are obtained,
(12)$$C_{j}(i)=\frac{1}{M}\sum_{q=1}^{M}c_{qj}(i)$$
(13)$$R(i)=\frac{1}{M}\sum_{q=1}^{M}R_{q}(i)$$

In Equation (12), *C_j_*(*i*) is the *j*th IMF component of x(i). If *M* is large enough, the sum of IMFs corresponding to white noise tends to zero. So, the original signal *x*(*i*) is decomposed as follows,
(14)$$x(i)=\sum_{j=1}^{k}C_{j}(i)+R(i)$$

In this paper, the amplitude of white noise is set as 0.1 PSI (Pounds per square inch, a unit of pressure) and *M* = 400.

## 3. Data Acquisition of Dynamic Pressure

The diagram of the data acquisition system is shown in Figure 3. In the system, the centrifugal compressor is driven by an 800 kW DC motor. The number of diffuser blades is 24, the number of impeller blades is 16, and the Mach number of the airflow in the pipeline is 0.6. Firstly, the outside air enters the pipeline through the intake filter, inner nozzle and chamber. After this, the filtered air is compressed by the centrifugal compressor. Finally, the compressed air is expelled into atmosphere through the exhaust muffler and outlet pipe by controlling the opening-degree of the two electric anti-surge valves with diameters of 250 mm and 100 mm. In the experiment, as the opening-degree of the anti-surge valves is decreased gradually, the centrifugal compressor will transit from a stable state to an unstable state. The dynamic pressure acquisition experiment lasts for 635.5 s, and the states of the centrifugal compressor changed from stable state to surge, and then to stable state again.

The acquisition system consists of three parts, namely the dynamic acquisition system, static acquisition system, and diagnosing and monitoring system. The virtual instrument made by National Instruments is used as the static acquisition system, its main function is to monitor the temperature of the machine, mass flow, inlet and outlet static pressure, and rotor velocity of the DC motor. The functions of the diagnosing and monitoring system are monitoring shaft vibration and axis displacement. As a dynamic acquisition system, CoCo80 manufactured by Crystal Instruments is used to measure the dynamic pressure at outlet. Moreover, the sampling frequency of the dynamic acquisition system is 20.48 kHz, and the unit of dynamic pressure is PSI.

In this study, the data from 200 s to 300 s are used, including the stable state, transition state, and surge state of the centrifugal compressor. The time sequence of the dynamic pressure is shown in Figure 4, and it is clear that with the anti-surge valve fully open, the centrifugal compressor operates in a stable state during the first 68 s. In this situation, the fluctuation of the dynamic pressure is faster and the amplitude of the waveform is smaller. With a decrease in the opening of the anti-surge valve, the fluctuation slows down and the amplitude of the waveform begins to increase gradually just after 68 s. In the interval between 68 s and 75 s, the system operates in a transition state, and surge occurs near to 75 s.

## 4. Multi-Scale Fuzzy Entropy Characteristics of Dynamic Pressure

### 4.1. Multi-Scale Fuzzy Entropy of Dynamic Pressure

The multi-scale fuzzy entropy curves based on the dynamic pressure with different scales are shown in Figure 5. In the diagram, the scale factor *τ* is set from 1 to 10, and the analysis unit is one second. From Figure 5, it can be seen that the curves of MFE begin to decrement at 68 s and tend to be stable at 75 s for all scale factors. The time point of the state change for MFE is consistent with that of the time domain waveform of dynamic pressure. Therefore, multi-scale fuzzy entropy can be used to describe the nonlinear characteristics of dynamic pressure. After studying the curves in Figure 5, it is clear that there are overlapping phenomena in the multi-scale fuzzy entropy curves under different scale factors, and the scale characteristics of the scale factor are not obvious.

In order to better visualize the scale characteristics of scale factors, the relationship between multi-scale fuzzy entropy and scale factor *τ* of the dynamic pressure is analyzed further, as shown in Figure 6. For this analysis, six time periods in different stages are used, that is, stable state (0–1 s, 40–41 s), transition (68–69 s, 70–71 s) and surge state (80–81 s, 90–91 s). From Figure 6, it can be seen that MFE differs greatly among the three stages, and the values of MFE are largest for the stable stage and smallest for the surge state at lower scale factors. The difference between the stable state and surge is significant for all scale factors. With the increase of the scale factor, the curves of the transition state intersect with the curves of the other two states. Consequently, it is somewhat difficult to distinguish the transition state from the other states of the system.

### 4.2. Improved Multi-Scale Fuzzy Entropy of Dynamic Pressure

In this section, the improved multi-scale fuzzy entropy of dynamic pressure is utilized in order to overcome the defects of the scale characteristics of MFE. The results obtained by IMFE are shown in Figure 7. Similarly, the scale factor *τ* is set from 1 to 10, and the analysis unit is one second. From Figure 7, we see that the tendencies of the curves of IMFE are consistent with those of MFE. Compared with the curves in Figure 5, the curves of IMFE show less overlap, and the differences between curves for different scale factors are seen more clearly. Therefore, IMFE is more suitable than MFE to analyze the nonlinear characteristics of the dynamic pressure.

Then, the relationship between IMFE and scale factor *τ* for the dynamic pressure at the outlet is analyzed. In Figure 8, the same six time periods as in Figure 6 are used. In contrast with the curves of MFE, the values of IMFE are largest for the stable stage and smallest for the surge state for all scale factors, the curves under the different states are smooth, the curves decrease with the increase of the scale factor for all three states and there is no intersection between curves for different states. Therefore, IMFE makes it easier to identify the different working states of a centrifugal compressor. Furthermore, it can be drawn that IMFE is a much more reliable solution to analyzing the characteristics of the centrifugal compressor in different states, because the values of entropies of the stable state and transition state are very close, as the scale factor is one.

In order to improve the anti-noise, EEMD combined with IMFE is used in this study. First, the time sequence of the dynamic pressure is decomposed by EEMD, and then the resulting IMF components are analyzed by IMFE in the following section.

## 5. IMFE Combined with EEMD of Dynamic Pressure

### 5.1. EEMD Decomposition of Dynamic Pressure

As shown in Figure 4, 100 s data including the stable, transition, and surge states of the centrifugal compressor are decomposed by EEMD. According to the definition and decomposition method of EEMD, the results of EEMD decomposition are not only related to the signal itself, but also depend closely on the amplitude and times of the white noise that is added after several experiments, the amplitude of white noise is set as 0.05 PSI. EEMD decomposition of the data is shown in Figure 9, including six IMF components and a residual component. From Figure 9, it can be seen that IMF1, IMF2, and IMF3 could not describe the change of the operating state of the centrifugal compressor. IMF4 could reflect the transition state, but it only weakly distinguishes the stable state from the surge state. Among them, IMF6 could clearly distinguish the stable state from the surge state; however, the transition state is not obvious for this component. As for IMF5, it appears to amplify fluctuation of the dynamic pressure, especially for the stable state.

### 5.2. Correlation between IMF Components and Dynamic Pressure

After EEMD decomposition, the correlation between IMF components and the original data is measured by the correlation coefficient in the frequency domain. Compared with time domain analysis, frequency domain analysis is more convenient, significant, and concise. The correlation coefficient *ρ* in the frequency domain can be expressed as follows [20],
(15)$$\rho =\left|\frac{E\left[\left(x(f)-u_{x}\right)\left(c_{j}(f)-u_{c_{j}}\right)\right]}{\sigma_{x}\sigma_{c_{j}}}\right|$$

In Equation (15), *x*(*f*) is the frequency component of *x*(*t*), *c_j_*(*f*) is the frequency component of the corresponding IMF, $u_{x}$ and $u_{c_{j}}$ are the mean values of *x*(*f*) and *c_j_*(*f*), respectively, and $\sigma_{x}$ and $\sigma_{c_{j}}$ are the variances of *x*(*f*) and *c_j_*(*f*), respectively, in the frequency domain.

The correlation coefficient ρ can be used to measure the correlation between each IMF component and the original signal in the frequency domain. If ρ = 0, the IMF component is uncorrelated with the original data; if ρ = 1, the corresponding IMF component is fully correlated with the original data exactly. Hence, a larger ρ means a larger correlation between the IMF component and the original data.

The correlation coefficients *ρ* of the IMF components, for the six time periods, representing three different stages, used in Figure 6 and Figure 8, are listed in Table 1. It can be seen from Table 1 that the correlation coefficients of IMF4, IMF5, and IMF6 are larger than those of the other three IMF components for these time periods. For IMF4, the value of ρ is the largest in the transition state. In conclusion, the results in Table 1 are consistent with those in Figure 9. The images in Figure 10, Figure 11 and Figure 12 are the spectrograms of the original data and the corresponding IMF4, IMF5, and IMF6 for the three time periods listed in Table 1. From these images, it can be seen that the maximums of the spectrograms for the original data and the IMFs at the stable stage are far lower and the other amplitudes are higher than those at the other two stages, which means that the system is in a stochastic state at the stable stage. The spectrogram of IMF4 does not indicate the characteristic of the stable stage clearly. At the surge stage, the frequency corresponding to the maximum amplitude of IMF4 matches with that of the original signal except for a small error. For IMF5 and IMF6, the curves of frequency spectrums match with the curve of the original signal at all three stages. In terms of the frequency and maximum amplitude, IMF5 matches well with the time sequence in transition, while IMF6 matches well at the surge stage.

### 5.3. IMFE of IMF Components for Dynamic Pressure

Because the characteristics of IMF4, IMF5, and IMF6 can clearly describe the working state of the centrifugal compressor in the frequency domain, the IMFE of the three IMF components are analyzed further in this section. Figure 13, Figure 14 and Figure 15 show the IMFE of the IMF components for the original time sequence of dynamic pressure. Part (a) of each figure shows the IMFE of the IMF components throughout the time period, and part (b) shows the IMFE of the IMF components under different scale factors of the six time intervals, namely, 0–1 s, 40–41 s, 68–69 s, 70–71 s, 80–81 s, and 90–91 s.

From Figure 13a, it can be seen that the curves of IMFE for IMF4 collapse at 68 s and IMFE increases sharply at 75 s. The values of IMFE are small for the time stage 68–75 s, as the centrifugal compressor operates in the transition state. From Figure 14a, it is clear that the variation of curves of IMFE for IMF5 in the stable state is larger than in the surge state, but the variation is not obvious for the transition state. In Figure 15a, the curves collapse at 75 s, and the changing of the curves of IMFE for IMF6 is contrary to that of IMFE for IMF4. As the system operates in surge state, the variation is very low and the values of IMFE are small for IMF6. As a result, it can be concluded that IMFE of IMF4 could exactly distinguish the transition state, and IMFE of IMF6 can be used to predict the occurrence of surge.

From part (b) of Figure 13, Figure 14 and Figure 15, similar to the IMFE of dynamic pressure in Figure 7 and Figure 8, the different working states of the centrifugal compressor can be identified clearly. Moreover, compared with Figure 8, the differences in IMFE of the IMF components are more obvious than the IMFE of dynamic pressure under most scale factors. Furthermore, it is easy to identify the transition state in the IMFE of IMF4, since the separation between stable state and unstable state is obvious. Similarly, the obvious separation occurs in transition, identified by the IMFE of IMF6. However, for the IMFE of IMF5, the separation occurs just before the transition, therefore the IMFE of IMF5 is more suitable to predict the surge than the IMFE of IMF6 in terms of dynamic pressure at the outlet.

Since each of IMFEs of IMF components can indicate a certain characteristic, it is significant to study the nonlinear characteristics of the dynamic pressure using IMFE of IMF components.

## 6. Statistical Reliability of IMFE of IMF Components for Dynamic Pressure

In order to test the statistical reliability of the IMFE of IMF components, the data of dynamic pressure are modified by adding randomly shuffled data. According the maximum absolute data of the dynamic pressure, white noise with an amplitude of 0.02 PSI is chosen as the randomly shuffled data. EEMD decomposition of the randomly shuffled dynamic pressures is shown in Figure 16, including six IMF components and a residual component *R*. The same conclusion is drawn from Figure 16 as from Figure 9, which infers that the noise has no influence on the analysis result obtained from the method.

Figure 17, Figure 18 and Figure 19 show IMFE of the IMF components for the shuffled dynamic pressure. Part (a) of each figures shows the IMFE of IMF components throughout the time period, and part (b) shows the comparison of the original and randomly shuffled dynamic pressure in IMFE of the IMF components, under different scale factors for the six time intervals, at 0–1 s, 40–41 s, 68–69 s, 70–71 s, 80–81 s, and 90–91 s. Moreover, the variation in the IMFE of IMF components shows properties consistent with the results for the original time sequence. In part (b) in Figure 17, Figure 18 and Figure 19, the curves also show consistent properties graphically with the curves of part (b) in Figure 13, Figure 14 and Figure 15, except for small errors.

## 7. Conclusions

In this study, as a powerful measure method for entropy, MFE, and IMFE of the dynamic pressure at the outlet of a centrifugal compressor was studied to obtain intrinsic properties. The results show that IMFE can exactly describe the complexity and the probability of the dynamic pressure under different scale factors. Specifically, the MFE and IMFE of the dynamic pressure decrease and become smooth as the system enters surge from the stable state, and the method of IMFE is more effective than that of MFE at certain scale factors. After studying the data, the results of the IMFE analysis of IMF components can provide detailed information regarding dynamic pressure in the frequency domain, because a slight fluctuation in the stable state and transition state could be revealed clearly in different IMF components. After analyzing the shuffled dynamic pressure, the results show that the method proposed in this study has certain anti-noise values and can be well-applied to the prediction and detection of surge for a centrifugal compressor. The subsequent work will use the results to accurately predict the initial surge of a centrifugal compressor, in order to prevent surge development.

## Figures and Tables

**Figure 1 entropy-22-00424-f001:**
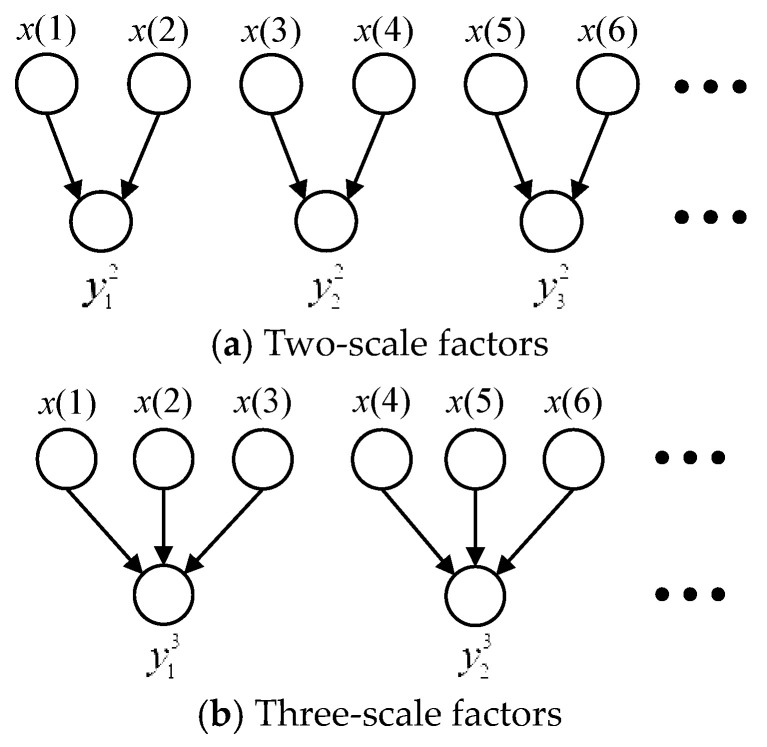
Coarse-grained data of multi-scale fuzzy entropy (MFE) with scale factors.

**Figure 2 entropy-22-00424-f002:**
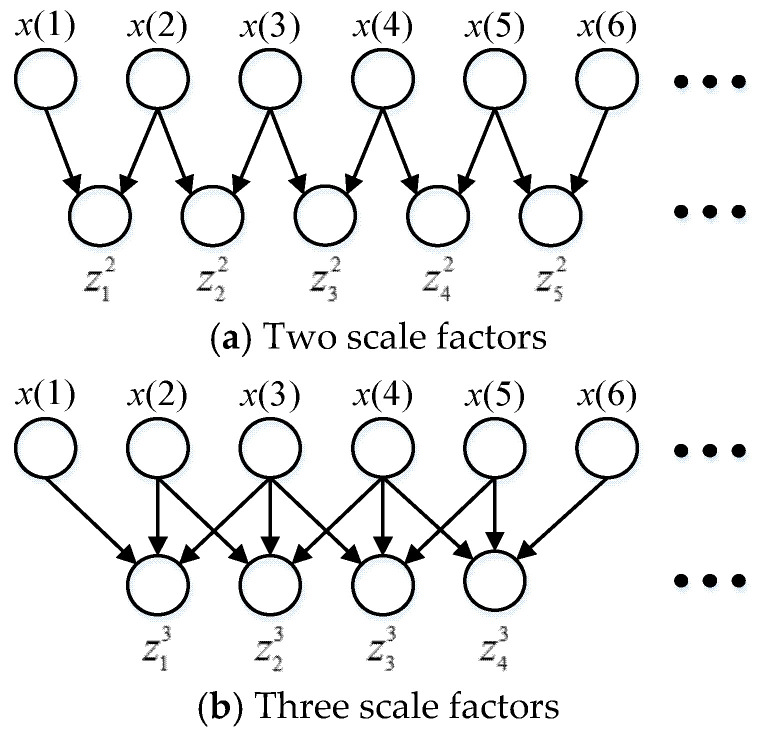
Coarse-grained data of improved multi-scale fuzzy entropy (IMFE) with scale factors.

**Figure 3 entropy-22-00424-f003:**
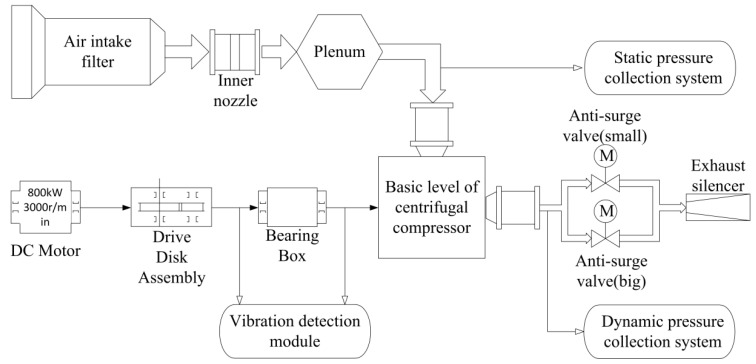
Data acquisition system.

**Figure 4 entropy-22-00424-f004:**
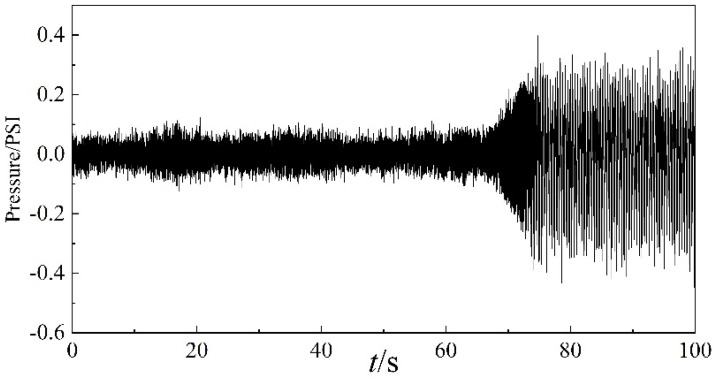
Time sequence of the dynamic pressure.

**Figure 5 entropy-22-00424-f005:**
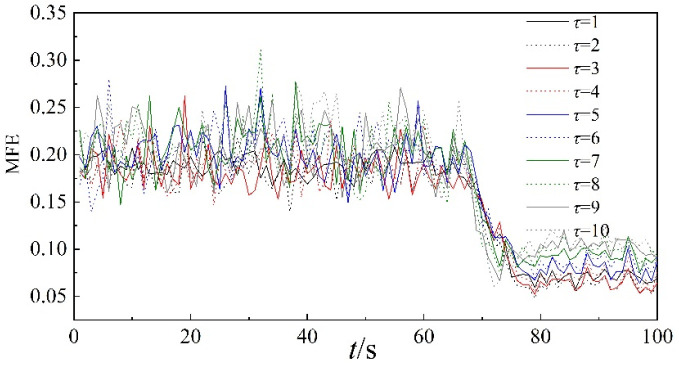
MFE of dynamic pressure of the whole time period.

**Figure 6 entropy-22-00424-f006:**
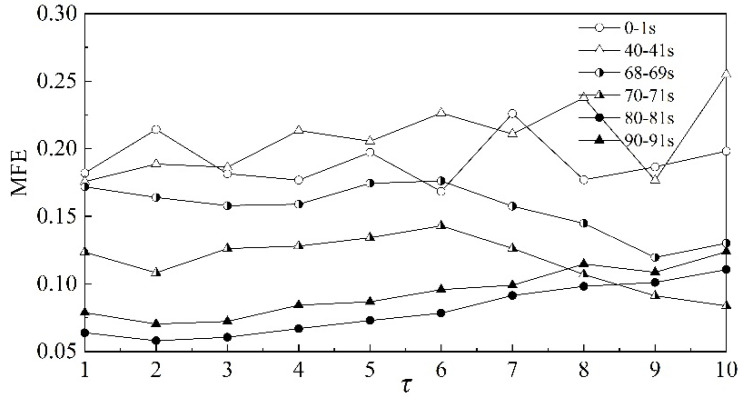
MFE of dynamic pressure under different scale factors.

**Figure 7 entropy-22-00424-f007:**
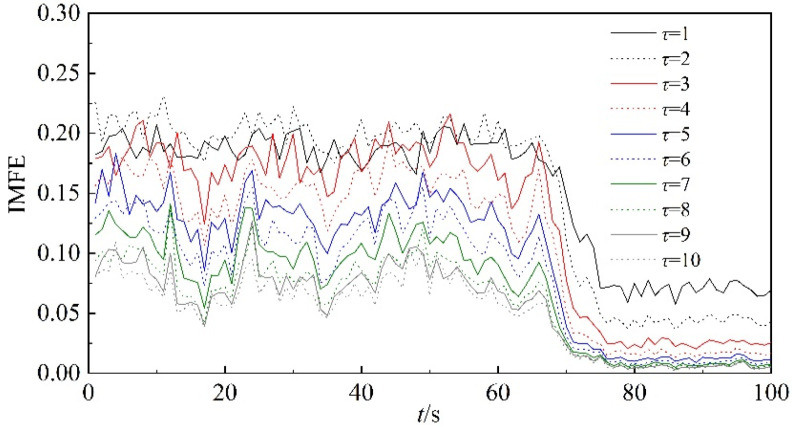
IMFE of dynamic pressure throughout the time period.

**Figure 8 entropy-22-00424-f008:**
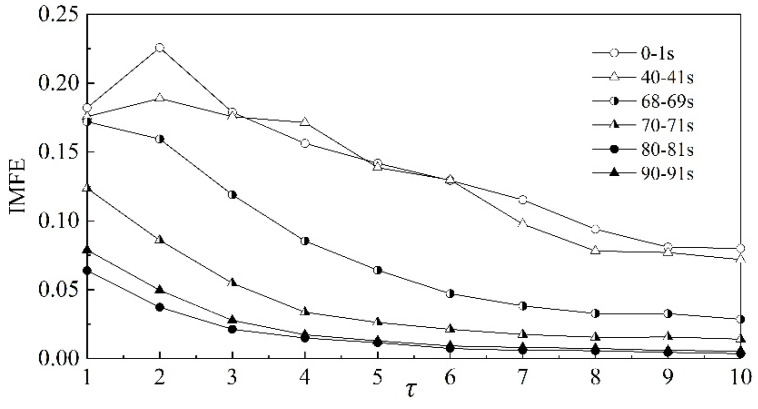
IMFE of dynamic pressure under different scale factors.

**Figure 9 entropy-22-00424-f009:**
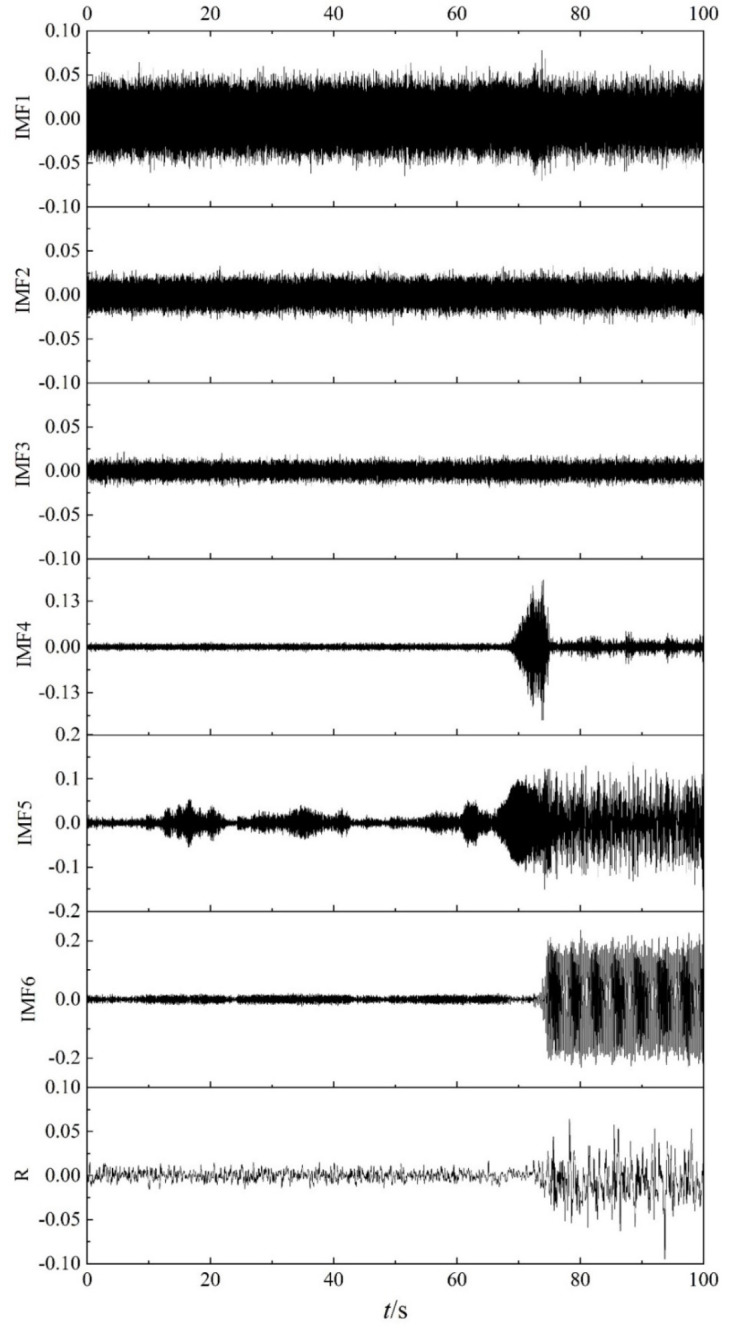
Ensemble empirical mode decomposition (EEMD) decomposition of dynamic pressure.

**Figure 10 entropy-22-00424-f010:**
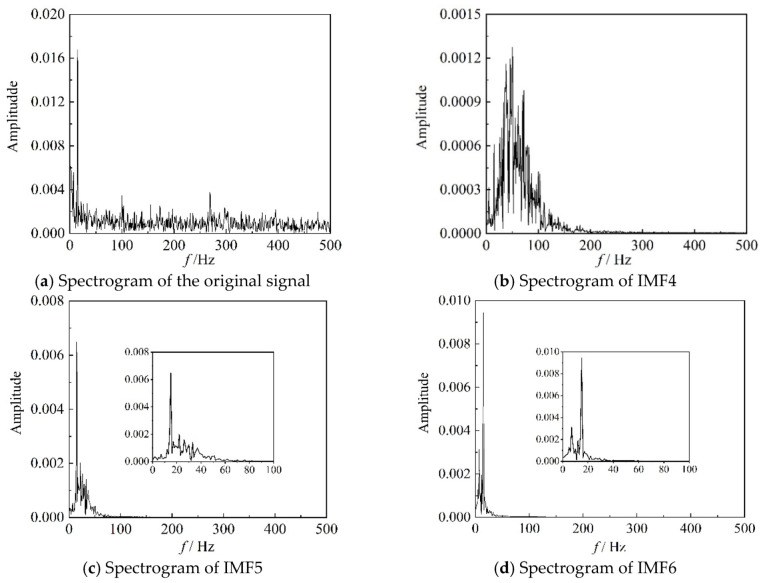
Spectrograms of original signal and better IMF components in 0–1 s.

**Figure 11 entropy-22-00424-f011:**
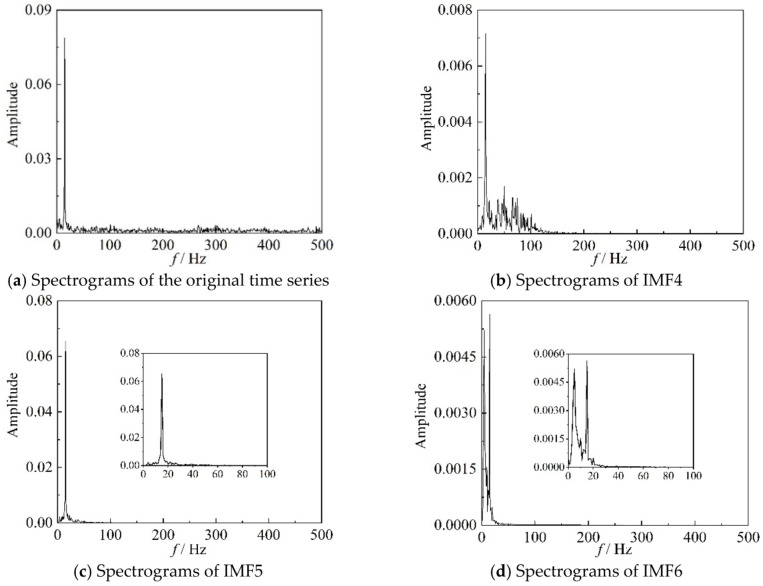
Spectrograms of original time series and better IMF components in 68–69 s.

**Figure 12 entropy-22-00424-f012:**
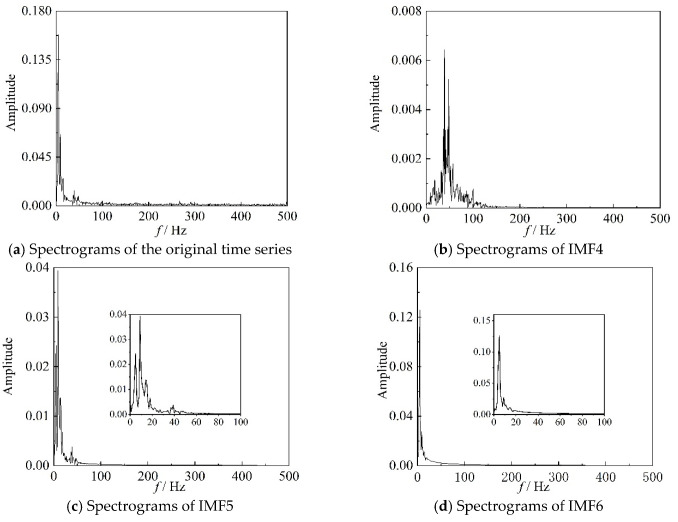
Spectrograms of original time series and better IMF components in 90–91 s.

**Figure 13 entropy-22-00424-f013:**
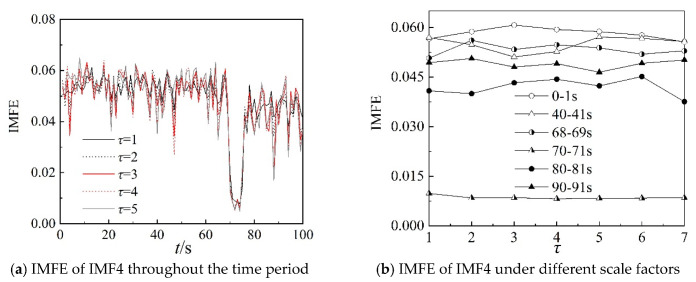
IMFE of IMF4 for original time sequence.

**Figure 14 entropy-22-00424-f014:**
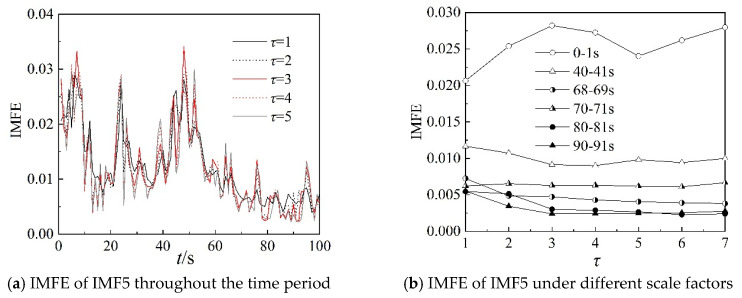
IMFE of IMF5 for original time sequence.

**Figure 15 entropy-22-00424-f015:**
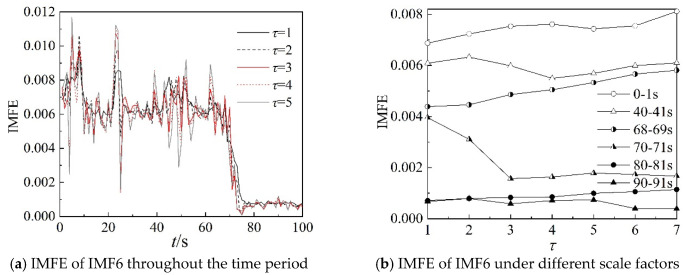
IMFE of IMF5 for original time sequence.

**Figure 16 entropy-22-00424-f016:**
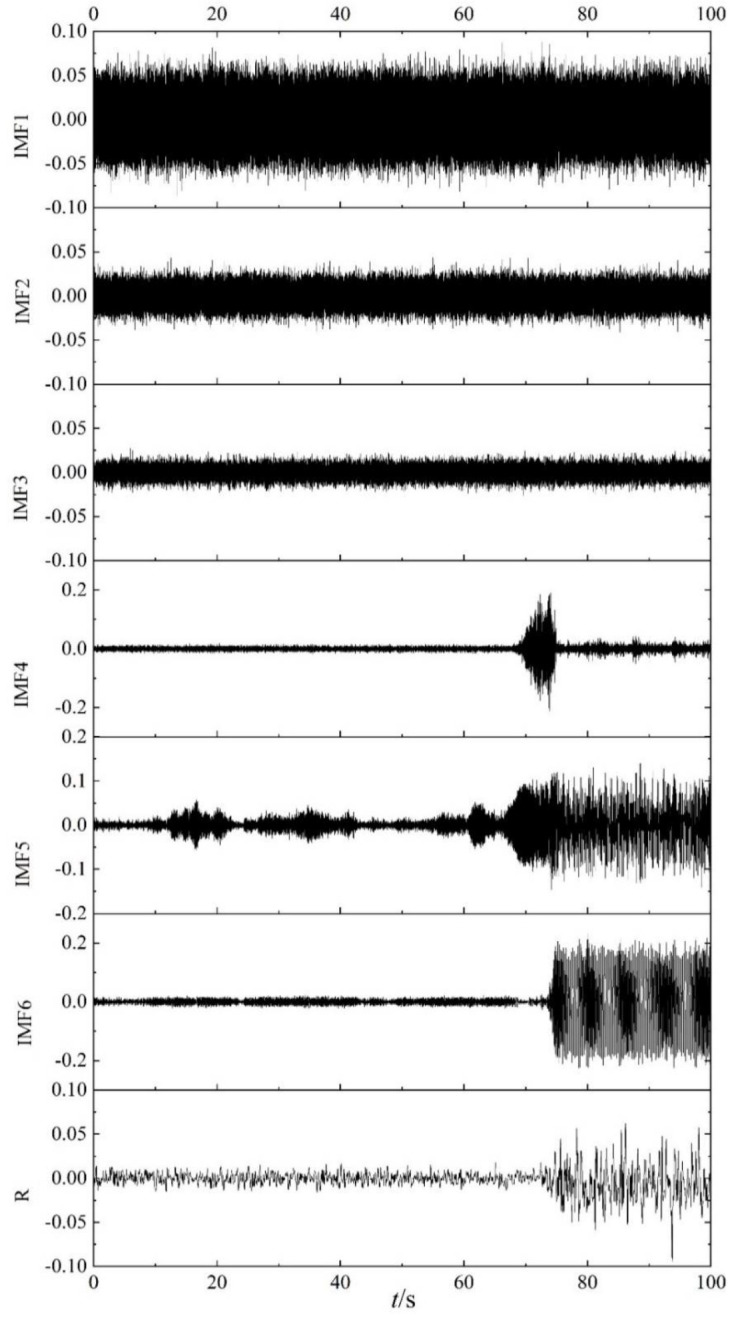
IMF components and residual component of shuffled flow pressure.

**Figure 17 entropy-22-00424-f017:**
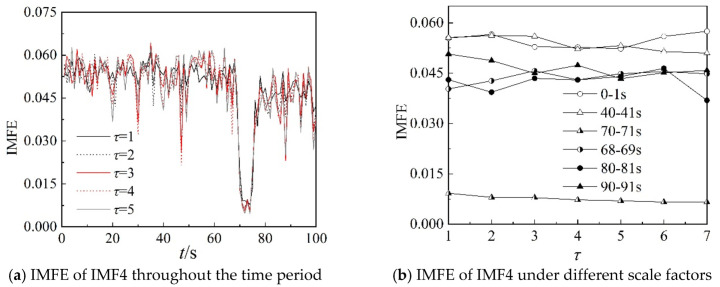
IMFE of IMF4 for shuffled time sequence.

**Figure 18 entropy-22-00424-f018:**
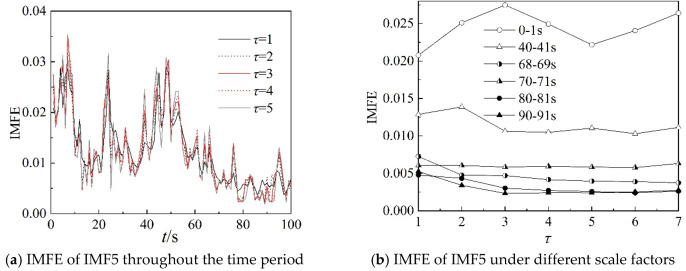
IMFE of IMF5 for shuffled time sequence.

**Figure 19 entropy-22-00424-f019:**
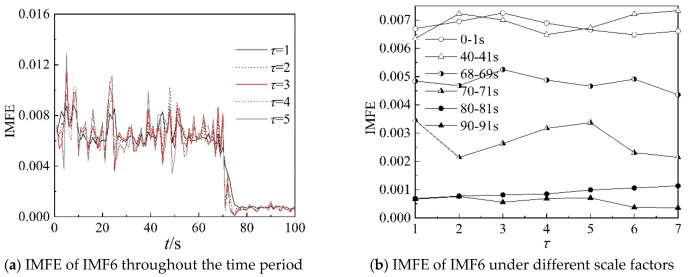
IMFE of IMF6 for shuffled time sequence.

**Table 1 entropy-22-00424-t001:** Correlation coefficients between the intrinsic mode function (IMF) components and the original signal.

Time Stage/s	ρ					
	IMF1	IMF2	IMF3	IMF4	IMF5	IMF6
0–1	0.2221	0.2708	0.2359	0.2789	0.6594	0.7137
40–41	0.1719	0.1821	0.1578	0.2715	0.8367	0.8609
68–69	0.0416	0.0808	0.1069	0.8078	0.9738	0.6558
70–71	0.0141	0.0340	0.0755	0.9824	0.9908	0.3345
80–81	0.1206	0.0277	0.0426	0.3183	0.7537	0.9467
90–91	0.1066	0.0117	0.0460	0.1701	0.8061	0.9786

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
