# Peer review of "Obtaining Information about Operation of Centrifugal Compressor from Pressure by Combining EEMD and IMFE"

_entropy, 2020, doi:10.3390/e22040424_

Round 1

Reviewer 1 Report

As a whole all suggestions provided in the review have been received. Some problems with grammar are still found. Enclosed a file with some errors to correct in the first two sections. Please try to control also in the following in the paper. However as the contents are concerned the paper is now acceptable.

Reviewer 2 Report

Dear Editors,

The manuscript was clearly improved. Nevertheless, the authors should improve English and correct several errors before publication. Some examples are given below, but more careful analysis is necessary. After the corrections are made, the manuscript may be published.

- On page 2, the second line, "it provides a way for studying surge in a sense" should be replaced by "it provides a way for studying surge" (delete "in a sense").

- "and routine methods, such as spectral analysis and time domain analysis, are fail to capture and describe these complex behaviors exactly" should change into "and routine methods, such as spectral analysis and time-domain analysis, fail to accurately capture and describe this complex behavior".

- "In this study, some fundamental theories in entropy ... are first introduced and developed" seems to be an overestimation.

- on line 96, X_k^m should be X_l^m;

- on line 98 there is a closing bracket missing;

- I think there is a mistake in Eq. (5). It is practically copied Eq. (4), without all the necessary adjustments. Actually, I think Eq. (5) is not necessary, since Eq. (4) is written for general m.

- There is a mistake in Eq. (7).

- What does it mean "MFE is not symmetric in its dependency on the samples of the original time sequence"?

- There are mistakes in Eq. (9).

- In line 170 the authors should write exactly what PSI means--not just "a gas pressure unit".

- line 181: "surg state"?

- line 196: "the fluctuation of _the_ dynamic pressure"

- line 260: "distinctly distinguish" -> "clearly distinguish"

- line 271: "means of" -> "mean values of"

- there is no need to add "in the frequency domain" in lines 271 and 272.

Author Response

This manuscript is a resubmission of an earlier submission. The following is a list of the peer review reports and author responses from that submission.

Round 1

Reviewer 1 Report

see pdf attached

Reviewer 2 Report

I have read the manuscript "Obtaining Entropy Characteristics of the Flow Pressure of Centrifugal Compressor ...", by Yan Liu, Kai Ma, Hao He and Kuan Gao. Although the concept and the motivation are interesting, the article is so poorly written and unclear that it cannot be accepted. I will give below some examples.

1. I guess Eqs. (3)-(6) refer to IMFE. This was not specified in the text, but it may be guessed from the forms of the equations. Nevertheless, in Eq. (6) the limits of the two summations seem to be wrong ("-τ+1" is missing from the upper limits).

2. Despite the fact that the function Φ was defined for IMFE, in Eq. (7) this is used to define the MFE. IMFE is not even defined, although it is used in Section 4.2.

3. The IMFE or MFE (whichever of these it may be used by the authors) depend on the whole set of values of the measured parameter. Then, it is plotted as a function of time in Figs. 5 and 7, without any explanations.

4. It is not clear (one may only try to guess, but nothing is presented and nothing is specified) on what sets of parameters are calculated the results presented in Figs. 6 and 8.

5. Several quantities are not defined, although they are used or even plotted. Among these quantities are EEMD and IMFs.

6. White noise and Gaussian noise are two different types of noise, not the same, as the authors write.

There are several other examples of presentation errors, but there is no point to try to make an exhaustive list. The language used is also poor, in general.

If the authors should rewrite the paper and resubmit.